# Coding Diagnoses from the Electronic Death Certificate with the 11th Revision of the International Statistical Classification of Diseases and Related Health Problems: An Exploratory Study from Germany

**DOI:** 10.3390/healthcare12121214

**Published:** 2024-06-18

**Authors:** Jürgen Stausberg, Ulrich Vogel

**Affiliations:** 1Institute for Medical Informatics, Biometry and Epidemiology (IMIBE), Faculty of Medicine, University Duisburg-Essen, 45122 Essen, Germany; 2Federal Institute for Drugs and Medical Devices, 53175 Bonn, Germany; ulrich.vogel@bfarm.de

**Keywords:** cause-of-death statistics, coding, death certificate, diagnosis, ICD-11

## Abstract

The 11th Revision of the International Statistical Classification of Diseases, Injuries, and Causes of Death (ICD-11) will replace its predecessor as international standard for cause-of death-statistics. The digitization of healthcare is a main motivation for its introduction. In parallel, the replacement of the paper-based death certificate with an electronic format is under evaluation. At the moment, the death certificate is used in paper-based format with ICD-10 for coding in Germany. To be prepared for the switch to ICD-11, the compatibility between ICD-11 and the electronic certificate should be assured. Objectives were to check the appropriateness of diagnosis-related information found on death certificates for an ICD-11 coding and to describe enhancements to the certificate’s structure needed to fully utilize the strengths of ICD-11. As part of an exploratory test of a respective application, information from 453 electronic death certificates were provided by one local health authority. From a sample of 200 certificates, 433 diagnosis texts were coded into the German version of ICD-11. The appropriateness of the results as well as the further requirements of ICD-11, particularly with regard to post-coordination, were checked. For 430 diagnosis texts, 649 ICD-11 codes were used. Three hundred and sixty two diagnosis texts were rated as appropriately represented through the coding result. Almost all certificates contained diagnosis texts that lacked details required by ICD-11 for a precise coding. The distribution of diseases was very similar between ICD-10 and ICD-11 coding. A few gaps in ICD-11 were identified. Information requested by ICD-11 for a mandatory post-coordination were almost entirely absent from the death certificates. The structure and content of the death certificate are currently not well prepared for an ICD-11 coding. Necessary information was frequently missing. The line-oriented structure of death certificates has to be supplemented with a more flexible approach. Then, the semantic knowledge base of ICD-11 should better guide the content related input fields of a future electronic death certificate.

## 1. Introduction

The World Health Assembly decided to accept and promote for implementation the 11th Revision of the International Statistical Classification of Diseases, Injuries, and Causes of Death (ICD-11) at the beginning of 2022, with transitional arrangements for at least five years [1,2]. By then, the member states of the World Health Organization (WHO) are requested to report their national cause-of-death statistics using ICD-11. Parallel to the introduction of ICD-11 [3,4], the replacement of the paper-based death certificate with an electronic one was also initiated by the WHO and national institutions (for example, in Australia [5]). Therefore, it is worth considering ICD-11 as well as ICD-10 in the implementation of the electronic death certificate.

Support of digitization was a major motivation for the further development of ICD with its 11th Revision. Table 1 lists some changes between ICD-10 and ICD-11, which aim at achieving this goal [6,7,8]. The conversion of ICD to an electronically implemented and ontologically founded infrastructure is intended to make ICD fit for the support of electronic charts and electronic records. But different revisions of the ICD pose a challenge for time series in cause-of-death statistics covering decades. At first sight, ICD-11 appears unfamiliar. Stem codes are primarily used for coding. With a few exceptions, stem codes can be used alone. Extension codes carry further characteristics of a condition and can be used in addition to stem codes [9]. Stem codes may have up to six digits with a dot after the fourth one. Numbers and letters are used for most of the digits. Each chapter of ICD now has its own character at the first digit of a code. Codes can be combined in clusters freely as well as following mandatory and optional rules. Clusters can consist of several stem codes as well as at least one stem code and extension codes.

In Germany, coding with ICD has a long tradition. Coding diagnoses of all inpatient and outpatient cases became mandatory with the ICD-10 in 2000 [10]. Germany switched from ICD-9 to ICD-10 in 1999 for the coding of causes of death. The statutory statistical offices of the federal states (the so called “Länder”) are in charge of cause-of-death coding with ICD-10 based on the information recorded with the sixteen Länder-specific death certificates. Standardized coding is ensured through training courses and through the use of the Iris software (cf. https://www.bfarm.de/EN/Code-systems/Collaboration-and-projects/Iris-Institute/_node.html (accessed on 13 June 2024)) in the Statistical Offices of the Länder. Because this coding is carried out subsequently to the processing of the death certificate, the switch to ICD-11 will not directly affect the documentation task of the people responsible for the postmortem examination. However, this documentation should provide the breadth and depth of information that is required for appropriate coding with ICD-11 to leverage the advantages of the new classification [11]. Current proposals of the death certificate’s electronic version rely on the paper-based version used with ICD-10 [12]. This study attempted to identify necessary changes in the electronic death certificate to support the subsequent use of ICD-11. This study will also contribute to a call for the development of a data set of ICD-11-coded death certificates [13].

This study did not intend to compare the reliability and validity of coding using ICD-11 versus ICD-10. ICD-10 was considered only to gain further insights into the demands of ICD-11 regarding the electronic death certificate.

## 2. Materials and Methods

### 2.1. Electronic Death Certificate

This study was part of a larger national project that that involved developing and piloting two Länder-specific statutory electronic death certificates. The German Federal Ministry of Health funded the project. The main project partners were the Federal Statistical Office (Destatis) and the Federal Institute for Drugs and Medical Devices (BfArM). Within this project, hospitals, general practitioners, and emergency physicians in two regions of Germany were equipped with the necessary equipment on a voluntary basis. The electronic death certificate was implemented as a progressive web app running on dedicated iPads provided by the Federal Statistical Office along with a printer to enable the production of a legally compliant format of the certificate. The equipment was reclaimed at the end of the project. The implementation of the electronic death certificate reflected the paper-based version to a large extent [12]. Diagnosis-related text fields of the Länder-specific paper-based versions were realized as free input options without plausibility checks. For each diagnosis-related text field, the option of recording an ICD-10-code was offered, comparable to the paper-based versions of the death certificates of the respective Federal States. Particularly for the two regions included, the physician responsible for the postmortem examination was obliged to input this code following the legislation of the respective Federal State. However, for the purpose of cause-of-death statistics in Germany, the free text information is coded by statutory statistical offices of the Federal States from scratch using the Iris automated coding system for causes of death [12]. Within the project, an approved electronic death certificate was automatically transferred to the responsible local health authority, checked, and then further transmitted to the statistical offices of the respective Federal State.

### 2.2. Data

For this study, one of the two local health authorities delivered data from 453 electronic death certificates collected between February 2023 and June 2023. The data included a selection of all available data elements. Data elements with identifiable information, for example, were excluded. Diagnoses could be assumed in 15 data elements, 4 covering the diseases or conditions that form part of the sequence of events leading directly to death (part 1 of the medical certificate of cause of death), 10 data elements containing conditions that do not belong to part 1 but whose presence contributed to death (part 2), and 1 explaining an external cause of death. For all 15 data elements, a free text field was supplemented with a field for ICD-10 coding and a field giving a time interval from onset to death.

In the data set, the 15 data elements contained 2220 entries representing 981 different character strings, at least one for each of the 453 certificates (cf. Figure 1). The spelling of the character strings was harmonized, ending up with 782 different diagnosis texts. Of the 2200 diagnostic texts, 2163 were coded with ICD-10 as the current process. To address the study objective of exploring the ICD-11 coding of electronic death certificates, a subsample of 200 certificates was selected for ICD-11 coding. This subsample included 31 certificates with at least one diagnosis text representing a “condition after” used in an interim evaluation, and 169 certificates randomly chosen from the remaining 422 ones. The selected 200 certificates covered 433 different diagnosis texts that were then coded using ICD-11.

The first author performed the coding with ICD-11. He first gained experience in diagnosis coding as assistant doctor using ICD-9. Then, he was involved in the management and quality of coding using ICD-10 in the fields of medical informatics, quality management, and health services research. At present, he is a member of the national ICD-11 working group. The coding was carried out in coordination with the second author, who is Head of Diagnosis Classifications at the Federal Institute for Drugs and Medical Devices (BfArM) and Head of the WHO Collaborating Centre for the Family of International Classifications in Germany.

The harmonized diagnosis texts were coded with the German draft version of ICD-11 for Mortality and Morbidity Statistics (ICD-11 MMS), using the coding tool and the browser provided by BfArM (cf. https://www.bfarm.de/DE/Kodiersysteme/Klassifikationen/ICD/ICD-11/uebersetzung/_node.html (accessed on 13 June 2024)). In cases of doubt, the respective tools of WHO were accessed.

ICD-11 and ICD-10 codes were formally checked against the metadata made available by the WHO as a single SimpleTabulation-file and BfArM as a collection of text files. For example, those metadata provided information about the position of a code (leaf versus node) or the precision of a code (residual versus non-residual). Only leaf codes are allowed for the coding of causes of death. Residual codes indicate missing information in the diagnosis text (not otherwise specified, NOS) on the one hand or a class that intentionally leaves details out (not elsewhere classified, NEC) on the other hand. With the exception of one ICD-10 code, all codes considered in the analyses were valid and terminal.

### 2.3. Analyses

The objective was split up into three research questions.

Are the certificate entries appropriate for coding with ICD-11?What differences exist between the coding with ICD-11 and ICD-10?What adaptations of the current electronic death certificate are needed to fully support cause-of-death statistics with ICD-11?

To answer the research questions, the coding of each diagnosis text was supplemented with a recording of additional information about the quality of its representation with ICD-11 (appropriate, not appropriate), about the availability and use of obligatory extension codes, and about the availability and use of optional extension codes. The data set and the coding results were maintained with the relational database management system Microsoft Access. Due to the exploratory nature of the study, only absolute and relative frequencies were calculated.

The study did not intend to compare the reliability and validity of coding using ICD-11 versus ICD-10. ICD-10 was considered only to gain further insights into the demands of ICD-11 regarding the electronic death certificate.

## 3. Results

### 3.1. Coding Results

Of the 433 diagnosis texts of the subsample of 200 electronic death certificates, 430 were coded with at least one stem code of ICD-11 (99.3%). In total, 649 codes were used with a mean of 1.5 codes per diagnosis text. Of the 649 codes, 494 were stem codes and 155 extension codes. Out of 433 diagnosis texts, 270 were coded with one stem code of ICD-11 alone (62.4%). For 124 out of 433 diagnosis texts, at least one extension code was used (28.6%). An extension code never occurred without a stem code. At a maximum, 10 codes were used to code a diagnosis text. The representation of the diagnoses text was rated as appropriate in 83.6% (*n* = 362). Only 1 out of the 200 electronic death certificates remained without any diagnosis text rated as appropriately coded with ICD-11. Out of 200 certificates, 129 included diagnosis texts that were all sufficiently coded (64.5%).

### 3.2. Appropriateness of Information in the Diagnosis Texts

Appropriateness was defined as availability of information in the diagnosis text necessary for a precise coding with ICD-11 (cf. Table 2). Of the 494 used stem codes of ICD-11, 226 represented a residual NOS-class (45.7%). In these cases, information might be missing in the diagnosis texts required by ICD-11 to assign a code representing a non-residual class. This affected 191 out of the 200 certificates of the subsample (95.5%). Of the 159 extension codes used, 44 represented such a residual class (27.7%). This affected 44 out of the 200 subsample certificates (22.0%). The proportion of residual classes was higher with ICD-11 than the result of 31.3% with the pre-coded ICD-10 entries (308 out of 983 codes in the sample). However, another 54 stem codes represented a residual NEC-class for a condition that could not be coded anywhere else (10.9%, ICD-10 4.6%). These conditions did not have a precise counterpart in the classifications. With ICD-11, extension codes were used to compensate this deficit in half of the diagnosis texts (*n* = 27).

With regard to the first stem code assigned to a diagnosis text, ICD-11 offered an optional post-coordination for nearly 80% and a mandatory post-coordination for about 20% (cf. Table 3). Out of the subsample, the optional post-coordination was at least partially used for 72 of the 433 diagnosis texts (19.0%), and the mandatory post-coordination for 6 diagnosis texts (1.4%). In most cases, information required for post-coordination with ICD-11 was missing from the diagnosis text.

### 3.3. Coding Differences between ICD-11 and ICD-10

The subsample of 200 certificates was coded with 983 ICD-10 codes and 1292 ICD-11 codes, of which 1068 were stem codes. Due to the structure of data recording, only one ICD-10 code could be assigned to a diagnosis text. For comparison, an average of 1.5 codes was used for coding with ICD-11.

Figure 2 depicts the distribution of ICD-10 codes and ICD-11 stem codes with regard to the aligned chapters of ICD. Only a few chapters present noticeable differences. The dominance of chapter 11 “Diseases of the circulatory system” decreased from a proportion of 34% with ICD-10 to a share of 28% with ICD-11. Codes of ICD-11 were more frequently used from chapters 01 “Certain infectious or parasitic diseases”, 08 “Diseases of the nervous system”, and 21 “Symptoms, signs or clinical findings, not elsewhere classified”.

The representation of 68 diagnosis texts from the subsample was rated as not appropriate with ICD-11, i.e., some information from these diagnosis texts got lost. The 68 diagnosis texts occurred in 90 entries from 88 electronic subsample death certificates. Coding with ICD-10 was rated as appropriate in 18 out of the 90 entries. The coding of ICD-10 and ICD-11 was compared for each of these 18 entries. Table 4 shows some exemplary explanations.

### 3.4. Adaptations Needed for ICD-11

ICD-11 requires a mandatory post-coordination for the first stem code of 83 of the 433 subsample diagnosis texts. The respective information for post-coordination was missing from 77 texts. One hundred and thirty four certificates were affected by the 77 texts with 212 entries. In each case, ICD-11 offers a selection of codes for the post-coordination of the causing condition (“has causing condition”). For 117 out of the 212 entries (55.2%), one causing condition could be identified in another entry of the same certificate. For 95 out of the 212 entries (44.8%), a causing condition could not be identified in any other entry of the same certificate.

Out of the 200 subsample certificates, 41 were affected by redundant coding with ICD-11, i.e., the multiple use of the same code (20.5%). Seven codes occurred three or four times on one or more certificates: 6D8Z “Dementia, unknown or unspecified cause” (four times); BA00.Z “Essential hypertension, unspecified” (four times); 5A11 “Type 2 diabetes mellitus” (three times); GC08.Z “Urinary tract infection, site and agent not specified” (three times); XT5R “Acute” (three times); XK8G “Left” (three times); and XK9K “Right” (three times). Out of the 200 subsample, 26 were affected by redundant coding with ICD-10 (13.0%). Five codes occurred three or four times on one or more certificates: F03 “Unspecified dementia” (four times); I10 “Essential (primary) hypertension” (four times); E11.7 “Type 2 diabetes mellitus with multiple complications” (three times); E11.8 “Type 2 diabetes mellitus with unspecified complications” (three times); and N39.0 “Urinary tract infection, site not specified” (three times).

## 4. Discussion

Our study confirmed that most death certificate diagnoses could be coded with ICD-11 [14]. Additionally, a dramatic increase in the number of codes was not seen that might be induced by ICD-11 features of post-coordination. Around two thirds of the diagnosis texts were coded with only one code. It should be mentioned here that the coding approach attempted not only at an assignment of a diagnosis text to a single class but also at a representation of the diagnosis text [15]. The frequent use of more than one code could therefore be expected. More than 80% of the coding results were rated as appropriate. This was an excellent result in light of the controversial discussion about the quality of the cause-of-death statistics in Germany [16,17].

However, the potential of coding with ICD-11 was not fully utilized. Information required by ICD-11 was missing from the diagnosis texts. On the one hand, residual NOS-classes of ICD-11 were frequently used with 45.7% (stem codes) and 27.7% (extension codes). In the respective diagnosis texts, information necessary for precise coding with ICD-11 will be missing. On the other hand, explicit recommendations for post-coordination could not be fulfilled in 76% (optional recommendations) and 93% (mandatory recommendation) of the diagnosis texts’ first stem code. This information might be missing from the respective diagnosis texts or be present in other parts of the death certificate. As a consequence, the training of the certifiers responsible for the postmortem examination regarding the information details requested by ICD-11 will be absolutely essential. Furthermore, due to the context-specific requirements of post-coordination, an active request for details should be embedded into the tool used for the recording of causes of death. The first research question of the study can be clearly answered. The certificate entries are currently not appropriate for coding with ICD-11.

One code per diagnosis text was used with ICD-10, whereas 1.5 codes were used on average per diagnosis text with ICD-11. This increase was to be expected due to the new features for post-coordination. From the point of view of longitudinal health surveillance, changes in the distribution of causes of death would be more important if this was caused by the introduction of a new version of ICD. With regard to the chapters of ICD, the distribution of ICD-10 codes and ICD-11 stem codes appeared quite similar. Some differences should be further analyzed in subsequent studies. In few cases, this study identified gaps in ICD-11 in comparison to ICD-10. Those gaps should be closed in the ongoing maintenance by the WHO. In answer to the second research question, no fundamental differences between ICD-11 and ICD-10 coding results were identified that would question longitudinal statistics covering periods with both ICD-10 and ICD-11.

The need to actively request information from the people responsible for the postmortem examination became very clear by analyzing the demands of ICD-11 for post-coordination. However, the requested information could be spread over several diagnosis-related entries on the death certificate. With regard to mandatory post-coordination, this was the case in more than half of the entries, which missed the requested information in their own diagnosis texts. Furthermore, coding single diagnosis texts independently from each other will lead to redundancies with regard to chronic diseases and supplementary extension codes. As a consequence, the answer to the third research question has to be affirmative, i.e., adaptations of the current electronic death certificate are needed to fully support cause-of-death statistics with ICD-11. The current line-oriented structure, with each line representing independent factors, has to be further developed to a matrix that takes into account overlaps and relationships between entries to a larger scale than that needed for coding with ICD-10.

Some limitations of this study have to be mentioned. The death certificates stem from one health authority. The coding of diagnosis texts with ICD-11 was carried out by a single expert. Therefore, the presented results should be confirmed on a broader scale, including a more diverse set of death certificates which are subsequently coded in parallel by two or more experts. At the moment, the results suffer from the exploratory character of the study. A consensus process with parallel coding by two or more persons would potentially increase the validity of the results. However, studies revealed a surprisingly high reliability of ICD-11 coding [18], substantially higher compared with ICD-10 coding [19]. There is therefore no reason to distrust the coding results of one experienced person. The available metadata of ICD-11 with the SimpleTabulation-file were limited. For example, information on mandatory or optional post-coordination had to be gained from the ICD-11 browser and ICD-11 coding tool. No metadata could be downloaded from the WHO that cover this information in a structured way for the complete ICD-11. Due to funding limitations, the WHO’s application programming interface (cf. https://icd.who.int/icdapi (accessed on 13 June 2024)) that offers web services to obtain programmatic access to ICD-11 could not be utilized to cope with this limitation. Rule-based mortality coding aiming to find the underlying cause of death was outside the scope of our study. A first study with 1248 death certificates revealed a lower accuracy with ICD-11 in comparison to ICD-10 [13].

## 5. Conclusions

To realize the potential of cause-of-death statistics with ICD-11, both the process of documentation by the certifiers responsible for the postmortem examination and functionality and layout of the death certificate must be further developed. Recording information needed by ICD-11 for correct diagnosis coding will require an active tool which guides the user through the death certificate. This tool will ask for related information and supplementary details defined in the terminological rules of ICD-11. The paper-based approach will fail to offer competitive support. The opportunities created by ICD-11 in a digitized healthcare system require digital solutions around ICD-11 itself. The layout of the form has to be redesigned to capture the complex relationship matrix between ICD-11 entries. Finding the appropriate design will be part of future work, ushering the electronic death certificate into the era of ICD-11.

## Figures and Tables

**Figure 1 healthcare-12-01214-f001:**
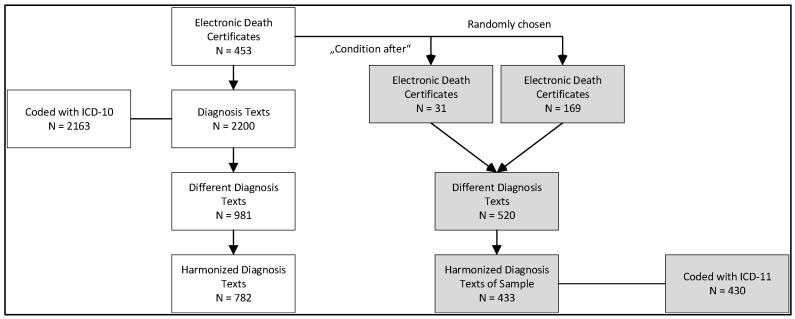
Selection of diagnosis texts. The data delivered by the local health authority already included ICD-10-codes (**left**). A total of 433 diagnosis texts were coded with ICD-11 within this study (**right**).

**Figure 2 healthcare-12-01214-f002:**
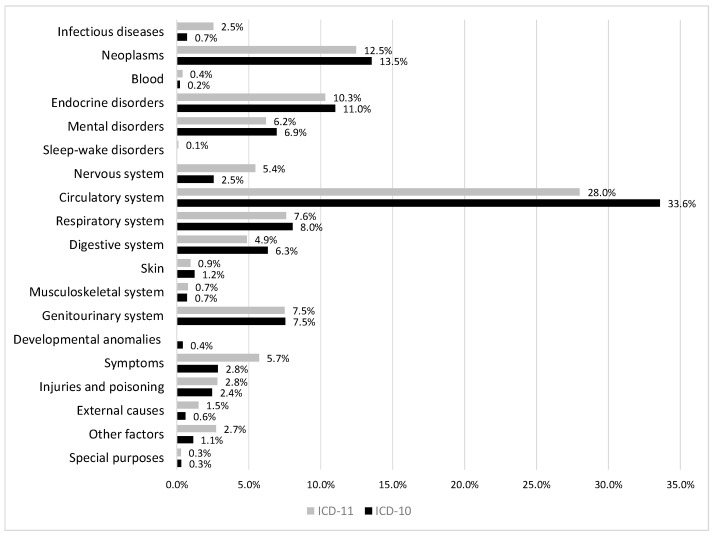
Distribution of ICD-10 codes and ICD-11 stem codes with regard to the chapters of ICD.

**Table 1 healthcare-12-01214-t001:** Advantages of ICD-11.

Criterion	ICD-10-WHO	ICD-11-WHO
Representation of current medical knowledge	Reluctant adaptation	Fundamental new setup
Terms	Outdated texts	Current use of language
Definitions	Chapter V	Comprehensive use
Structure	Limited to four-digit codes with 10 options at a maximum	Six-digit stem codes with extended character set
Hierarchy	Tree structure	Polyhierarchy (multiple parenting)
Post-coordination	Star-asterisk system	Clustering
mandatory		“code also”
optional		“use additional code”
Metadata about diagnoses	Not covered	Covered by extension codes
Terminological foundation	Not available	Integrated
Use case	Paper-based favored, electronically possible	Electronically

**Table 2 healthcare-12-01214-t002:** Use of residual classes.

	ICD-11 Subsample, *n* (%)		ICD-10 Sample, *n* (%)
	Stem Codes	Extension Codes	
Not otherwise specified (NOS)	226 (45.7)	44 (22.0)	308 (31.3)
Not elsewhere classified (NEC)	54 (10.9)	0 (0.0)	45 (4.6)
Total	494 (100.0)	159 (100.0)	983 (100.0)

**Table 3 healthcare-12-01214-t003:** Frequency of diagnosis texts with regard to post-coordination.

Post-Coordination	Optional, *n* (%)	Mandatory, *n* (%)
Not applicable	3 (0.7)	3 (0.7)
Offered, not used	261 (60.3)	77 (17.8)
Offered, partially used	67 (15.5)	0 (0.0)
Offered, completely used	15 (3.5)	6 (1.4)
Not offered	87 (20.1)	347 (80.1)
Total	433 (100.0)	433 (100.0)

**Table 4 healthcare-12-01214-t004:** Exemplary diagnosis texts that could not be appropriately coded with ICD-11.

Diagnosis Text	ICD-10	ICD-11	Comment (Number of Entries)
Alcoholic cirrhosis of liver	K70.3	DB94.10 Alcoholic hepatitis with cirrhosis DB94.3 Alcoholic cirrhosis of liver without hepatitis	The status (with or without) of hepatitis must be known in ICD-11 (*n* = 2).
Stage 3 of a chronic kidney disease	N18.3	GB61.2 Chronic kidney disease, stage 3a GB61.3 Chronic kidney disease, stage 3b	ICD-11 requires a distinction between stage 3a and 3b (*n* = 8).
Perforation of the intestine	H63.1	XA9607 Gastrointestinal tract XA6452 Small intestine XA1B13 Large intestine	The topography of the intestine is missing in ICD-11, either pre-coordinated or using an extension code. ICD-11 only offers a distinction between the small and large intestine as the first split in the hierarchy below gastrointestinal tract (*n* = 1).

## Data Availability

Restrictions apply to the availability of these data. Data were obtained from a local health authority and are available from the authors with the permission of this local health authority.

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
