# Peer review of "Coding Diagnoses from the Electronic Death Certificate with the 11th Revision of the International Statistical Classification of Diseases and Related Health Problems: An Exploratory Study from Germany"

_healthcare, 2024, doi:10.3390/healthcare12121214_

Round 1
Reviewer 1 Report
Comments and Suggestions for Authors
Coding diagnoses from the electronic death certificate with ICD-11 an exploratory study from Germany
1. In Table 1, what do you mean (3.1) and (4.2). Please make it clarified.
2. There is repeated information in Table 1.
3. The gap, previous studies on ICD-11, and contribution of this study should be more highlighted in Introduction. There are many pilot studies in different countries.
4. Who coded the diagnoses using ICD-11? Her/his training in ICD-11? How many people coded the diagnoses?
5. Did you apply mortality coding rules (SP1-SP8 and modification rules) for each certificate? or were diagnoses coded without considering other diagnoses in a certificate?
6. How can we interpret figure 1 when ICD-11 had 1.5 code per diagnoses but ICD-10 had 1. These differences are normal, I think.
7. How did you evaluate the shift of codes between chapters? There are codes (diseased) have been moved from a chapter to another chapter. How did you consider this in figure 1 and comparing ICD-10 and ICD-11?
8. There are studies compared ICD-10 and ICD-11 in terms of accuracy and reliability. They could be cited in discussion for further interpretation of the results. For example:
· Comparison of the accuracy of inpatient morbidity coding with ICD-11 and ICD-10. doi.org/10.1177/18333583231185
· Eisele A, Dereskewitz C, Oberhauser C, et al. (2019) Reliability, usability and applicability of the ICD-11 beta draft focusing on hand injuries and diseases: results from German field testing. International Journal for Quality in Health Care31(10): G174–G179.
· Impact of the ICD-11 on the accuracy of clinical coding in Korea. doi:10.1177/18333583221095147
Comments on the Quality of English Language
-
Author Response
See file with our point-by-point responses.

Reviewer 2 Report
Comments and Suggestions for Authors
Paper Review: Coding diagnoses from the electronic death certificate with ICD-11: an exploratory study from Germany
Summary: This study delves into the potential of ICD-11 and ICD-10 codes for coding electronic death certificates. A total of 453 electronic death certificates were coded in both ICD-11 and ICD-10. The study aimed to assess the suitability of the certificate entries for ICD-11 coding, identify deviations between ICD-11 and ICD-10 in coding, and propose necessary changes to the current electronic death certificate to enhance cause-of-death statistics through ICD-11. Encouragingly, over 80% of the coding results were deemed satisfactory. The authors also observed that ICD-10 codes were chosen 1.5 times more frequently than ICD-11 codes on average. To fully harness the potential of ICD-11 in electronic death certificate coding, they concluded that it is imperative to enhance documentation processes by postmodern examination certifiers and revamp the layout and functionality of the electronic death certificate.
Article: The hypotheses are highly testable and could be stated even more clearly as the authors refer back to their research questions in the conclusion as confirmed or not by their study.
The weakest section is the Methods section, which requires more description of the study design (testing both ICD-10 and 11 or just ICD-11?). See further notes below. The methods were also mixed into the results section and need to be separated. The results would be better presented with more tables, as the wording is hard to follow.
Review: The topic is relevant and warrants further exploration. However, the clarity of the article could be enhanced. As per the title, the initial aim is to examine ICD-11 coding for electronic death certificate coding, yet both ICD-11 and ICD-10 are tested. It was initially unclear if ICD coding had been previously used for electronic death certificate coding. Thus, the methods section lacks key information about current coding practices, who codes, and which ICD is used for death certificates. The research objectives are robust and clear. However, the changes required in terms of documentation and layout could be more specific regarding how and where electronic death certificates should be enhanced.
Specific Comments:
Title: Consider changing the title to ‘Dual coding diagnoses from the electronic death certificate with ICD-10 and ICD-11: An Exploratory Study from Germany. (make sure to capitalize after the colon)
Abstract: Abstracts are easier to read with Background, Methods, Results and Conclusion section headings
Line 10: Consider changing at least as to “…as the international standard…”. Remove ‘at least’
Line 11: Consider changing health care to “health care data collection” for clarity.
Line 14: Consider a wording change from “…has to be make sure” to “…the compatibility between ICD-11 and the electronic certificate must be assured.”
Line 14: Consider changing “Objective…” to “The first objective was…”, or ‘A primary objective was…”
Line 15: Consider changing the word elaborate to the word describe.
Line 22-23: The sentence “Nearly any certificate was concerned with….” Should be reworded for clarity.
Line 27: Consider changing “The line-oriented structure has to be…” to “The line-oriented structure of death certificates has to be…”
Line 28-29: The sentence “Then, the terminological knowledge behind ICD-11…” needs to be reworded for clarity. Terminological knowledge is not common language or ICD-11 language.
Line 33: Consider changing “…put into force…” to “…accept and promote for implementation…” to make the sentence sound less casual.
Line 37-40: More clarity is needed if an ICD version is already used or not in Germany and what are the standards for data collection – by whom? How educated?
Line 42: Please describe upfront why ICD-10 is relevant in this study and who codes, along with how ICD-11 is coded.
Line 45: The sentence “This might contract with the need…” could be reworded for clarity.
Table 1: The table seems to present conflicting information since ICD-10 seems to be favourable, this comes back to the clarity on if the goal is to implement ICD coding for death certificates in general or if the plan is to upgrade to ICD-11. Who replaced the paper-based death certificate? Was it the WHO, or did individual countries choose to? There is duplicate data in the table as ‘Representation of current medical knowledge’ is repeated along with the content below and should be removed due to redundancy. Insert 2 spaces minimum under the table before the text.
Line 61-62: What is meant by “…in times of…”? Does it mean while ICD-10 is in use or where ICD-10 is in use? Consider rewording for clarity.
Line 63: Consider defining what ‘it’ is. The wording should be changed to “This study will also…”
Line 64: Consider changing the word claim to call. “…a call for the development of a data set…”
Line 67: Consider changing the wording from “…project developing and testing…” to “…project that involves developing, testing, and implementing statutory death certificates.”
Line 72: Consider changing the word infrastructure to the word equipment. Or be specific about what is included in the infrastructure.
Line 75: Consider changing the word infrastructure to the word equipment.
Line 77: For “extent.8” Is the number 8 a typo or reference?
Line 79: Why an ICD-10 code? This study is about ICD-11. Please clarify if ICD-10 is currently used for death certificates in Germany and provide more rationale.
Line 85: Again for “death.8” Is the number 8 a typo or reference?
Line 91: Please state the beginning and end dates of data collection not just the end date.
Line 92-93: The sentence “The provided data set contained…” seems out of place. Consider stating the inclusion and exclusion criteria for data collection which would be clearer, or the specific elements collected and not collected.
Line 99: ICD-10 coding or ICD-11?
Figure 1: Still some confusion on the use of ICD-10 being new or not.
Line 109: Consider changing “…into ICD-11…” to “…using ICD-11…”
Line 114: Clarify what metadata is being referred to. Is it a gold standard? What does it consist of?
Line 119: Please describe what a deviation might be in the methods above – as to why both ICD-10 and ICD-11 are being used. Consider calling it dual coding for comparison purposes. If so, justify why dual coding is necessary and useful in this study.
Line 123: What is meant by the word recording? Is it a voice recording? Documentation? Further data collection? Who did the coding and what training did they receive for both ICD-10 and ICD-11 coding? More elaboration is needed regarding these methods.
Line 137: The sentence “The representation…” could be changed for clarity. More context for who and how this data was collected is needed before the results will make sense.
Line 143: The definitions need to be in the Methods section, only results should be in the Results section.
Line 145-147: Please explain residual and non-residual. The reader needs more context and definitions regarding these terms.
Line 149-150: Consider grouping all the comparative results (ICD-10 vs. ICD-11) in a separate paragraph or section. Inserting it here confuses the reader. What is meant by “These shares were higher…”? Do you mean the proportion was higher or the ratio? Consider being more specific.
Line 150: Are you referring to ICD-11 in this section? The section is not very clear. Use of a table or figure may be helpful.
Line 159: ICD-11 doesn’t ’request information’ – anthropomorphism. Consider saying “information required for post-coordination within ICD-11 was missing…”
Line 164: If the stem codes were unique, consider adding “unique stem codes”; if it was only an absolute count, leave the wording.
Line 168: Consider changing “Only few…” to “Only a few…”
Line 169: Consider rewording “…decreased from a share of…” for clarity. Perhaps the word proportion would be best.
Line 175-177: What is meant by not appropriate? These definitions need clarity. The word comes up in lines 175-176 and 177.
Line 179: A table showing ICD-10 codes and ICD-11 codes then a comments box with a description of the issue or difference would be clearer for the reader. It is hard to follow the meaning or examples in this format.
Line 182: Is the code N18.3 an ICD-10 or ICD-11 code.
Line 183: Consider changing “ICD-11 demands a…” to “ICD-11 requires a…”
Line 190: Consider changing “ICD-11 requested a mandatory…” to “ICD-11 requires a mandatory…”
Line 200: Consider rewording “…fourfold at least one certificate…” for clarity.
Line 205: Reword for clarity.
Line 209: This section is not very clear. Consider a table for comparisons.
Line 211: “Our study confirmed the promising coding results with ICD-11…” is a vague statement. Consider stating your strongest result at the onset of the discussion section. An example of possible wording is “Our study confirmed that most death certificate diagnoses could be coded using ICD-11…”
Line 221: What is the reversed perspective?
Line 233: After the word ‘answered’, consider using a semicolon or period. A connecting phrase could also be used. An example is “such that the certificate entries…”
Line 235: “Having one…” is incorrect grammar. Consider rewording for clarity.
Line 249-251: Should post-coordination be mandatory? Is that level of specificity needed about deaths, or is a higher-level diagnosis sufficient? Can stem and post-coordination codes be reliably replicated between people coding the same record? Does it introduce too many options and the potential for differences in code choice? What is the recommendation for the WHO?
Line 260-261: The Methods section needs to describe the expert's training for ICD-11 coding.
Line 264-265: How is reliability measured if there is only 1 coder? Define what is meant here.
Line 267: What is meant by metadata?
Line 269: What application programming interface? ICD-Fit? Is it part of the equipment? Consider adding more description to the Methods section about this.
Comments on the Quality of English Language
It was evident that English may not be the authors' primary language. Some terms and phrases could be improved to make them clearer for the reader.
Author Response

(The authors gave the same response as above.)

Round 2
Reviewer 2 Report
Comments and Suggestions for Authors
Healthcare-Electronic Death Certificates & ICD-11
Review #2
Thank you for addressing the majority of the comments. The paper is becoming clearer. We have a few further comments that need to be addressed.
1. Abstract: A statement must be made that death certificates in Germany are currently coded using ICD-10. Simply saying 'it makes sense to switch from ICD-10 to ICD-11..." is not enough to establish that ICD-10 is standard practice already.
2. The statement in the introduction (Lines 40-41) “Therefore, it is worth considering ICD-11 as well as ICD-10 in the implementation of the electronic death certificate.” is very vague and does not explain the specific roles of ICD-10 and ICD-11 in this study. See other comments. This lack of clarity is our main concern with this paper.
3. Section 2.2. Data needs more clarity.
a. First, a screenshot or mock-up of the form might be useful to accompany your description of the electronic death certificate, as I was lost in your text description.
b. Line 107: “In the data set,…” should be the start of a new paragraph because now you are describing your randomization and study data set/flow sheet vs the data contained in the electronic form.
c. Line 109: “The spelling of the character strings was aligned …” Aligned with what? Do you mean assigned? The meaning is not clear.
d. After the word ‘texts.’ On Line 110, a sentence that better describes the role of ICD-10 coding in the study should be included. For example: “Of the 2,200 diagnostic texts, 2163 were previously coded with the ICD-10 as part of the current process.
e. Likewise, following that ICD-10 statement, a statement such as “To address the study objective of exploring the ICD-11 coding of electronic death certificates, a subsample of 200 certificates was selected for ICD-11 coding. "For some analysis" is too vague. Be specific about why this group was partitioned off.
f. In addition, the last line of that paragraph could be made more specific by adding, “The selected 200 certificates covered 433 different diagnosis texts that were coded using ICD-11”.
g. Figure 1, on the left-hand side, says, “Diagnosis Texts n=2.200.” I believe this should be 2,200.
h. Figure 1, in the far right-hand box, add n=433 to be clear and to match the left-hand side box describing the ICD-10 coded texts that indicates n=2163.
i. The Figure 1 caption more aptly describes the method – information that should be better described in the body of the methods section.
j. Line 117 - Which codes? You haven't described how the ICD-11 codes were assigned - who did it? I see it below - but any process related to the coding should be explained before assigned codes are described. See the following notes for rearranging this paragraph.
k. On page 4 of 9, From Line 125 to the end of the paragraph, place this information about the coder and his experience at the beginning of this paragraph. This still says very little about training but it is acceptable. Then, add the information (lines 117-124) about how codes were assigned below it.
l. Line 125: Poor word choices. Change “He made his first experiences..” to "he gained experience in diagnosis coding…"
4. Section 2.3 Analysis – point 2 - use the word 'differences.’ The word
‘Deviations’ is out of place here.
5. Line 145 – “The study did not intend to compare reliability and validity of coding using ICD-11 versus ICD-10. ICD-10 was considered only to gain further insights into the demands of ICD-11 regarding the electronic death certificate.” This is a good statement. A statement similar to this needs to be added at the end of the introduction or the start of the methods section – restating this in multiple places would make it much clearer.

The English is much improved.
Author Response
Thanks to the reviewer again for the helpful remarks. We considered the recommendations nearly complete. There is only one exception. Unfortunately, we do not have a screenshot of the application.